# Statistical Review of Quality Parameters of Blue-Green Infrastructure Elements Important in Mitigating the Effect of the Urban Heat Island in the Temperate Climate (C) Zone

**DOI:** 10.3390/ijerph17197093

**Published:** 2020-09-28

**Authors:** Patryk Antoszewski, Dariusz Świerk, Michał Krzyżaniak

**Affiliations:** Department of Landscape Architecture, Poznań University of Life Sciences, Dąbrowskiego 159 Street, 60-594 Poznań, Poland; patryk.j.antoszewski@gmail.com (P.A.); dariusz.swierk@up.poznan.pl (D.Ś.)

**Keywords:** urban heat island mitigation strategy, blue-green infrastructure effectivity factors, blue-green infrastructure parameterization

## Abstract

Urban Heat Island (UHI) effect relates to the occurrence of a positive heat balance, compared to suburban and extra-urban areas in a high degree of urbanized cities. It is necessary to develop effective UHI prevention and mitigation strategies, one of which is blue-green infrastructure (BGI). Most research work comparing impact of BGI parameters on UHI mitigation is based on data measured in different climate zones. This makes the implication of nature-based solutions difficult in cities with different climate zones due to the differences in the vegetation time of plants. The aim of our research was to select the most statistically significant quality parameters of BGI elements in terms of preventing UHI. The normative four-step data delimitation procedure in systematic reviews related to UHI literature was used, and temperate climate (C) zone was determined as the UHI crisis area. As a result of delimitation, 173 publications qualified for literature review were obtained (488 rejected). We prepared a detailed literature data analysis and the CVA model—a canonical variation of Fisher’s linear discriminant analysis (LDA). Our research has indicated that the BGI object parameters are essential for UHI mitigation, which are the following: area of water objects and green areas, street greenery leaf size (LAI), green roofs hydration degree, and green walls location. Data obtained from the statistical analysis will be used to create the dynamic BGI modeling algorithm, which is the main goal of the series of articles in the future.

## 1. Introduction

Cities from around the world face the same problem of overheating. Experiments confirm that this phenomenon occurs in over 400 cities [1,2,3,4]. However, due to the etiology of the phenomenon, it can be thought to occur in every human settlement. The UHI effect is caused directly by local climate change, which is a consequence of changes in land cover, i.e., urbanization, and indirectly by global climate changes. Due to the negative impact of these factors on biodiversity and quality of life in the city, mitigation of UHI is probably one of the most significant challenges of the early XXI century [1,2,3,4].

Strategies and plans have been introduced to mitigate the UHI effect [5]. Specific solutions that fit into these activities can be divided into the material, structural, and planning strategies. These are: (1) the use of building and finishing materials with a high solar reflection coefficient, (2) the use of evaporation systems on roofs and roads, (3) the introduction of water facilities, green areas and green technical solutions, (4) reduction of anthropogenic heat emission and energy consumption, (5) urban planning taking the local climate into account, and (6) modifying the topography of a city [6,7,8,9,10,11,12,13,14,15,16,17,18,19,20,21].

Many researchers show that the most effective mitigation solution for UHI is to introduce and increase the area of already existing urban greenery, water facilities, and create the so-called Blue-Green Infrastructure (BGI) [22,23,24,25].

The scientific problem of recent times is to find one universal model for creating the BGI focused on UHI mitigation. However, there are certain uncontrollable variables, and it is quite impossible to develop a holistic method for creating green infrastructure for all cities around the world. One of these uncontrolled variables is climate. This is a serious obstacle when we try to suggest nature-based solutions developed by scientists. The problem is, for example, the different distribution of annual temperatures, the difference in the daily temperature distribution, wind distribution, and the percentage of humidity between individual climates, etc., which often cause a different effectiveness of BGI solutions in the context of UHI mitigation [1,2,3,4]. These differences become problematic in creating a model for general use and specifying design guidelines.

To create the most valuable holistic model, one should limit the technology transfer to solutions used in one climate zone. The temperate climate zone is characterized by a stable connection with the increased intensity of the UHI effect and, at the same time, an extensive BGI solution knowledge base. Understanding the value of the parameters responsible for the quality of these solutions is crucial for the development of BGIs in cities. The urban designers’ community is the target recipient group of solutions developed by the scientific community dealing with urban climate changes. The lack of clear systematization of data relating to the modulable BGI parameters specified for UHI mitigation is a necessary gap to be filled. The data will allow scientists to choose settings for BGI elements in the existing context in the future and create BGI most efficiently from a mitigation of elevated temperature viewpoint.

This paper aims to systematically present geometric, structural, and spatial parameters of BGI elements that may affect mitigation of the effect of the UHI in a temperate climate zone. This paper is also the first in a series of articles describing the creation of a dynamic algorithm for thermal optimization of cities in the temperate climate zone using BGI solutions. The lack of a clear systematization of the BGI-related design parameters specified for the mitigation of UHI is a gap that needs to be filled. In the future this may help to select the most appropriate parameters of BGI elements and to create green infrastructure in the most efficient way in terms of UHI mitigation quality.

### 1.1. Urban Heat Island (UHI) Effect

The UHI effect is the best-documented phenomenon of climate change [5]. This phenomenon was first studied around 100 years ago [26,27,28]. UHI refers to the occurrence of a positive heat balance in a city, compared to suburban and rural areas [5,29,30,31,32,33]. UHI intensity is measured by readings of air temperature at a meteorological station located in an urbanized area and at a reference station located outside an urbanized area. Increased temperature can be seen in the winter at night or before dawn (anthropogenic heat emission). The same characteristic is also visible during sunny days in the summer, when urbanized artificial surfaces absorb more solar energy and are warmer than those in rural areas characterized by a high degree of surface shading and high humidity [31]. The amplitude of UHI can fluctuate around 5 °C and varies depending on the characteristics of the cities, weather conditions, the method used to measure the temperature distribution, and the selection of a reference station [1], as well as the occurrence of extreme weather events like heat waves.

The UHI is present in all cities of all climate zones on Earth [22]. Still, its intensity is appropriate for each human settlement [14] and increases as the level of urbanization increases [22]. Therefore, UHI is the primary product of urbanization [34].

The growing population of cities [35] directly affects the housing demand, which causes constant development of construction [14]. The increase in urbanized areas is associated with significant changes in land cover, primarily through the disappearance of green spaces and natural biologically active areas [25]. Urban areas significantly differ in terms of heat capacity from rural spaces without buildings [14]. Until now, solar energy was considered to be the main factor of UHI [36]. However, sources indicate that there are other far more affecting processes. Some elements of this mechanism have been identified as uncontrolled [37], including climatic, meteorological, and geographical characteristics [38,39,40]. There are also controllable factors [37] associated with decision-making processes [22,39], which are shown in Table 1.

The latter parameters (Table 1) related to natural elements are among the most frequently mentioned in the literature. In addition to the factors mentioned above that directly affect the occurrence of UHI, global warming generated by greenhouse gases has played a unique role in shaping the future climate of cities [26].

It is essential to understand the full impact of UHI as a set of consequences for the environment and society [47,57]. The effects of UHI are deterioration of internal and external thermal comfort, exposure to heat stress, increase in cooling energy consumption, energy peaks, rise in electricity charges, the concentration of poisonous pollutants, increase in intensity and frequency of heat waves, increase in greenhouse gas emissions, change of local wind patterns, increasing humidity, changes in rainfall rates causing thunderstorms and floods, and changes in local ecosystems [2,14,22,24,31,58,59,60,61,62]. There is evidence that the UHI effect increases mortality and morbidity [63] due to urban populations being pushed beyond their adaptability [25]. Diseases such as damage to the thermoregulatory system caused by heat stress, cardiovascular stress, thermal exhaustion, heatstroke, and cardiovascular diseases cause thousands of deaths annually and can be directly caused by UHI [48,62,64].

UHI is a significant research problem because it indirectly accelerates global warming by significantly increasing energy consumption per person (20 W), as well as global temperature (1 °C) [65,66]. Therefore, it is necessary to develop effective prevention and mitigation strategies for UHI.

### 1.2. Blue-Green Infrastructure (BGI)

The concept of BGI originated in the United States in the 1990s [67], but the term was first used in an urban context in Europe in 2006 [68,69]. BGI can be defined as a strategic network of planned green areas, covering both the public and private places, and managed as an integrated system providing several environmental benefits [25,67]. The functional meaning of BGI is the same as the so-called gray infrastructure, but with natural materials as its matter. The blue infrastructure includes lakes, water reservoirs, rivers, wetlands, swamps, water engineering facilities, rain gardens, etc. The green infrastructure includes forests, arable fields, non-urbanized areas, grasslands, woodlands, lawns, parks, private gardens, sports facilities, green roofs, green walls, and other facilities [22,25,70,71,72,73]. BGI is a key element in cities, providing a range of ecosystem services such as reducing air pollution and improving the health conditions of urban residents [20,74]. Besides, BGI primarily mitigates the effects of heat islands by affecting the thermal environment of the city, which is confirmed by extensive literature [22,72,73,75,76,77,78,79,80,81]

It is well known that plants provide a cooling effect and can, therefore, be used in strategic urban and architectural projects [22,82,83,84]. The introduction of urban systems and green areas is perceived as the most cost-effective UHI mitigation strategy, while also taking into account ecological safety [85], which is why BGI is widely used worldwide [86,87]. BGI can reduce the intensity of UHI by regulating surface energy exchange processes through heat dispersion, evapotranspiration, regulation of emissivity, and influencing air movement and heat exchange [25,27,69,72,73,88,89]. The heat capacity of water is much higher than that of air. Vegetation directly absorbs light and heat, which is then dissipated as latent heat [22,72,73], which clearly reduces heat stress [90]. By increasing the amount of greenery in the city or developing green infrastructure, one works directly on the source of the problem by reducing extremes and fluctuating air and surface temperature. New urban projects should provide the introduction and reintroduction of greenery to the built-up urban environment.

To properly design green infrastructure aimed at UHI mitigation, however, one must understand the relationship between land cover patterns and Land Surface Temperature (LST) [91,92]. It has been accepted that urban greenery cools its immediate surroundings [93], and the essential BGI factor in mitigating the impact of UHI is the percentage of plant coverage. However, many other structural, geometrical, and topographical elements influence the quality of UHI mitigation [76,92,94]. These factors include, among others, distance of the area from the nearest BGI element, edge density and shape complexity, structure of the plant compound, material structure of the BZI pavement (water, greenery), and distance from the neighboring BGI element—synergy effect [92,94,95,96,97]. Therefore, UHI can be softened not only by balancing the relative amounts of different land cover features, but also by optimizing their configuration [92].

## 2. Materials and Methods

In the XXI century, the profession of urban space designer is increasingly combining scientific knowledge with design practice. To correctly design nature-based solutions in the urban environment, it is necessary to rely on the latest scientific achievements. It is particularly important in the case of BGI design focused on increasing the quality of the urban environment. Not many review articles exist on the systematizing of the quality parameters of BGI elements (Table 2). Besides, only some of them cover all BGI solutions. Although these articles focus on the effectiveness of UHI mitigation by a given BGI solution, they mainly focus on uncontrolled parameters. This approach limits the possibility of transferring information about the most important parameters from the urban planners’ point of view, i.e., transfer of controllable parameters of BGI elements to the field of design practice, where this knowledge is most needed.

Another problem for urban planners and designers may be a too-vague approach to data collection. Climate is one of the main uncontrolled factors affecting the intensity of the UHI phenomenon; it also substantially affects the effectiveness of BGI solutions [98,123]. Most review papers, however, are based on data from different climate zones, which often do not allow for the transfer of knowledge and the use of advanced solutions in cities with different climates. Pérez et al. [118] point out that the classification of research results should take place within one climate zone to increase their normativism. Hence in obtaining data for this paper, it was decided that we focus on only one climate zone.

UHI intensity is related to the increase in urbanization, which is why areas with a high density of human settlements become at risk of overheating. Besides, high UHI intensity occurs in large and “finite” cities, which are characterized by inhibiting population growth. The highest densities of human settlements are found in East Asia, Europe, and North America [124]. After applying the Koeppen-Geiger climate classification map to the base map, the coverage of these critical areas with the warm temperate climate zone (C) area is visible. The situation is similar in the case of cities with a low population growth rate. These cities are also in most cases in the warm temperate climate zone [35]. This theory can be confirmed by [125]. Cities with the highest UHI intensity are also located to a greater extent in a warm temperate climate. Analyzing the above information allows us to conclude that it is the warm temperate climate zone that is associated with the highest frequency occurrence and highest intensity of UHI in the world. Hence, the most reasonable choice is to select temperate climate (C) zone as the criterion for choosing test results.

### 2.1. Selection Criteria for Papers

The main objective of the bibliographic query was to reduce the number of review articles on BGI elements targeting UHI mitigation. Only articles describing case studies from the temperate warm climate zone and comparing the quality parameters of a given BGI element were considered.

In our research, we used the normative procedure used in systematic reviews related to UHI papers [126,127]. The process consists of four stages: (1) identify broad search criteria to obtain a universe of research; (2) limit the research universe to eligible literature based on stringent and clear criteria; (3) collect information from qualified documents and encode them into statistical data; and (4) present discussions on the results of selected studies.

We conducted a literature query in recognizable databases: Journal Citation Reports (JCR), GreenFILE/EBSCO, ELSEVIER/ScienceDirect, CAB ABSTRACTS, Arts and Humanities Citation Index (AHCI), AGRICOLA–AGRICultural OnLine Access, Academic Search Complete, MasterFILE Premier, Science Citation Index Expanded (SCIE), Scopus, Social Sciences Citation Index (SSCI), Springer/Kluwer, Web of Science, and Wiley-Blackwell.

Development of strategies for effective relevant papers search turned out to be the problem, due to the lack of uniformity in the name of research related to the BGI. After an in-depth analysis, the “Green Infrastructure” query was selected as a nomenclature adopted in the scientific community in the world. To avoid using outdated research, we focused on the last twenty years. The result of the query was 661 publications. The papers were then checked for suitability for testing. Review and non-research papers were rejected at this stage. The result of the first phase of delimitation designated 452 papers for the detailed phase of eligibility checks. This phase qualified the usefulness of the publication in two stages. In the first stage, studies carried out in a climate zone other than temperate climate (C) zone, studies carried out in more than one climate zone, and studies for which location was not specified were rejected, resulting in the number of articles being limited to 266. The second stage involved selecting only articles describing and comparing the geometrical, structural, and spatial parameters of BGI components. As a result of this delimitation process, 173 publications qualified for the final research.

The logical diagram for selecting papers for review is presented in Figure 1.

### 2.2. Statistical Analysis of the Results

To prepare the result matrix and interpolate the literature review data into numerical data, a zero-one distribution was applied. Each publication was treated as 1 n, and then a matrix was constructed that contained repeatable variables regarding:blue-green infrastructure elements: water structures, green areas, greenery along the streets, green roofs, and green walls;parameter family: geometrical and morphological parameters;country of origin of the publication.

Described above action allowed for the use of an advanced statistical technique for this type of descriptive data.

Statistical analyses were based on the discriminatory analysis. The result of the analysis was to determine which of the parameter families are most relevant to BGI and in which countries they were most often tested. Canonical variate analysis (CVA)—a canonical variation of Fisher’s linear discriminant analysis (LDA)—was used to construct the model.

Ordinance techniques were used, ordering research trials along a gradient represented by both the ordinate and abscissa axis. The Compliance Analysis (CA) was performed to check which of the techniques would be the most appropriate for the analyzed dataset. This procedure aimed to answer the nature of the structure of the analyzed data set based on the gradient length (linear or unimodal). The gradient length (>3) suggests that the CCA (Canonical Conformity Analysis) is appropriate for this type of dataset.

The discriminant analysis compared parameter families and their testing frequency. We also checked which parameter families are most relevant for BGI. For this purpose, the progressive stepwise analysis was used. All variables were assessed. The variables that most contributed to group discrimination (based on the *p* and F values for each analyzed variable) were included in the model. This process was repeated until the *p*-value fell below 0.05 for the examined variable. The Monte Carlo permutation test was performed to determine the significance level (separately for each variable and then for the entire model). All tests, calculations, and graphic elements were prepared in the Canoco for Windows software and Microsoft Excel spreadsheet. The following tools were used from the Canoco for Windows software: Canoco for Windows 4.5, CanoDraw for Windows, and WCanoIMP.

## 3. Results and Discussion

A significant reduction of papers in the last phase of delimitation resulted from the fact that the researchers often analyzed the impact of only one parameter on UHI mitigation (e.g., only one configuration of a green roof). This resulted in a lack of qualitative comparisons in a group of parameters. Appendix A presents the results of our research. It contains geometrical, structural, and topographic parameters (depending on the spatial context) of BGI elements cited in scientific papers as influencing UHI mitigation. The significance of individual parameters was expressed in percentages, depending on the number of papers (Table 3).

### 3.1. Discussion and Recommendations for Urban Design Strategies

The results of the analysis (Table 3) showed that research on blue-green infrastructure geared to mitigate UHI in a warm temperate climate zone focuses on the role of green areas in cities. The fewest articles concerned research on the development of blue infrastructure. The least citations were related to the group of geometric parameters, which can be explained by the relatively small range of this group of parameters. Morphological parameters were most often cited in the literature. They are essential when creating green areas, green roofs, and walls. Particular attention should be paid to the building material of these objects and their configuration. Topographic parameters turn out to be important when developing green infrastructure in the urban canyons. It seems logical that the urban canyon, as a specific place significantly affecting the intensity of UHI, has the most substantial impact on the efficiency of the BGI structure itself.

The CCA analysis showed that green walls, street greenery, and green areas, in general, were the most important objects for BGI in the analyzed literature as indicated by the remarkably high F values in Table 4. For green areas, green roofs and walls, morphological parameters proved to be important, depending on the characteristics of the object itself, while for street greenery and water facilities, topographic parameters related to spatial aspects were important (Table 4). The distribution of publications due to the country of origin is very dispersed. However, some dependencies can be found. In German, Portuguese, Israeli, Hungarian, and Swiss papers, the most described parameters related to green areas and greenery along streets. In France, Spain, Serbia, or Greece, the topic of green roofs and walls was most often discussed. The enormous variety was found in publications from China, Hong Kong, and the USA (Figure 2). In addition, the largest number of publications was recorded in Hong Kong, China, and the United Kingdom, respectively. The number of new landscape infrastructure elements may be influenced by green policies established in these countries such as “The Hong Kong Greening Masterplan” for Hong Kong, “Sponge Cities” for China and “Green Infrastructure Planning Practice Guidance” for the UK [128,129,130].

In addition to the essential parameters of BGI elements from the UHI mitigation point of view, there are those less critical, but still having a lowering effect on temperature. Those BGI structures are presented below, and the impact of their parameters on the ambient temperature is characterized. This section is intended to help city designers understand the operation of cooling island mechanisms and recognize the importance of conscious BGI design as a remedy for climate change.

### 3.2. Water Structures

Urban Blue Infrastructure (UBI), i.e., open surface waters, is an essential element of the urban environment responsible for many ecosystem services whose UHI mitigation functions are often overlooked [81,131,132]. Analyses show that water objects can lower the air temperature in their surroundings by up to 2.5 °C [81]. Evaporative cooling plays an important role in the relief of elevated temperatures by UBI, in which absorbed solar energy in the form of short-wave radiation is converted into latent heat by producing steam. At the same time, the surface energy of water can be transferred by conduction and convection. The thermal properties of water (specific heat capacity and enthalpy of evaporation) give it high thermal inertia, modeling the air temperature during the day and acting as a thermal buffer [133,134,135,136]. The UBI includes all types of water structures, both natural and artificial, including stagnant water reservoirs such as lakes and ponds, but also wetlands and shorelines, and running water reservoirs such as rivers, streams and canals, fountains, waterfalls and cascades, and water walls, as well as elements of sustainable water management [81,103]. However, it should be noted that the impact of water facilities on UHI mitigation is seasonal, especially in watercourses. In the spring and summer, water reservoirs cool down the surroundings, but in autumn and winter, they influence the ambient temperature slightly [137].

For water objects, the most frequently cited parameter affecting the temperature reduction was the area of the BGI structure. Such a result meets the accepted opinion that the size of the BGI object is the most critical parameter affecting the reduction of the ambient temperature [135,138]. It appears that large lakes have a strong cooling effect, and this effect can be observed far away from the site [135]. Studies show that one large water body has a more significant impact on UHI mitigation than several smaller water bodies of regular shape with the same volume of water [134]. However, several smaller lakes affect a larger area of the city [135,139]. Hence, their effectiveness should be seen synergistically and qualitatively rather than in aggregate. Size matters for rivers—the wider they are, the more significant and stable the cooling effect is [140].

Deepwater reservoirs have a more significant impact on mitigating the ambient temperature compared to shallow reservoirs [141,142]. Theeuwes et al. [135] argue that deep reservoirs with well-mixed water bring more significant thermal effects than reservoirs in which water is not intensively mixed. However, in the case of increased UHI intensity, even shallow water objects can cool the temperature in their surroundings [141]. This ability is due to the possibility of thermal exchange with the entire water column. The problem in this case, however, is fast evaporation—the reservoir must have a constant water supply in order not to dry out [142]. Due to the limited free space in the city and for security reasons, it is suggested to create structures composed of shallow (small) water reservoirs, ponds, and water gardens—in particular as part of a sustainable drainage system for urban areas [141,142].

The shape of water reservoirs is also an important factor affecting the intensity of cooling [138]. The shape of the reservoirs should be relatively simple. Water reservoirs with a quadrilateral or circular shape have a more significant impact on cooling the surroundings [140,143]. A more complicated shape leads to lower L_max_ and T_max_ values [143]; research conducted in Copenhagen confirms this argument. When the size of the BGI element is limited (<1 ha) and its shape is compact (square or round), it is more effective in reducing LST than a larger reservoir (>1 ha) of complex shape [139].

The heat absorption by a water reservoir depends directly on its temperature and ambient temperature, which is why the immediate vicinity of the reservoir or river is important. Covering the surroundings of blue infrastructure with impervious surfaces has a negative effect. They cause a lower capacity efficiency of the reservoir due to a higher temperature. Therefore, it is necessary to increase the share of greenery in the river or lake surroundings and to reduce the share of impermeable surfaces [143,144].

The effect of water on temperature drop decreases inversely in proportion to the distance from a lake. However, the effect of the lake on temperature is still noticeable a few kilometers away [135]. The range of the cold island created by reservoirs is strongly influenced by wind speed and direction. Therefore, it is necessary to correctly place the BGI object. It should consider the complexity of the wind flow around buildings. The geometry of buildings and their arrangement is also important, both from the windward and leeward of reservoirs [136]. Therefore, the location of water reservoirs in urban tissue is crucial [135].

### 3.3. Green Areas

Urban greenery such as grassy areas (with rare or absent trees), parks, sports facilities, and urban forests can provide a cool island in the urban structure [131,132,137,145,146].

In the case of urban green areas, the most frequently cited parameter was the area of the BGI object. The size of green areas is positively correlated with the intensity of cooling. This relationship is not linear [147], and the effect of a cool island becomes significant when the park’s surface exceeds 14 km^2^ [148]. Although the shape of a park is not as important as its size, studies show that it can also have a significant impact on the intensity of a cool island [149]. Lu et al. [149] present the round shape as ideal, and studies confirm this by showing that irregular and elongated parks perform worse compared to regular, compact parks [148].

The volume of greenery in the park also matters. The greater it is, the greater the cooling effect [150]. This phenomenon becomes noticeable only after exceeding the 69% threshold of green cover. In addition, parks with a total green area of less than 10,566.25 m^2^ have little impact in an ambient temperature decrease. When the green area is more than 740,000 m^2^, the distance the cold air reaches grows very slowly [151]. However, a park can become an island of heat if the impervious surfaces reach more than 50% [76,152]. The results of Zhou and Cao [153] also showed that the size of the plant cover is more important than the spatial distribution when determining LST.

The species structure of the green area is an important quality parameter. The presence of trees is a factor that increases the thermal quality of such space, which is associated with a decrease in the sky visibility factor (SVF) [22,150,154]. Higher levels of a shadow cast by trees and bushes (+50%) can contribute to an ambient temperature reduction of 0.2–0.4 °C [154]. Research results of Lee et al. [155] and Duncan et al. [156] show that trees and shrubs are more effective than grasslands at mitigating UHI. Tall and broad deciduous trees growing in a compact group ensure maximum cooling in parks [157,158,159]. However, it is shrub vegetation that can be used in any urban green area [156]. The very complexity of the plant community can affect the quality of cooling. The various species structure and a significant number of species, in a way due to a larger shading area, can have a positive effect on the thermal environment [160].

The impact of irrigation systems on temperature reduction is noticeable in many papers [103,146,161,162,163]. Irrigation can reduce the temperature in the park during the heat waves. However, after exceeding a certain threshold, irrigation becomes less effective in alleviating elevated temperatures. The results of the analysis carried out by Broadbent et al. [164] show the effect of irrigation time on the temperature during the day. It turns out that night irrigation brings the most satisfactory results.

The nature of the buildings surrounding the greenery plays an important role in the cooling effect. Green areas show the greatest UHI mitigation effect in highly urbanized space [103,165]. In the cases studied by Perini and Magliocco [90], it was proved that plants near buildings in a high building density area provided better effects in the relief of UHI. There is a connection between the cooling effect and strength and direction of winds. The best location of green areas is the area located higher on the wind flow line than the UHI critical area [22,166,167,168]. The distance from the park logarithmically reduces the possibility of cooling [94]. Studies show that the cold island buffer is usually less than 500 m, which is caused by thermal disturbances [95,152]. However, there are mentions that the cold island does not spread further than 30 m, which is probably due to the variability of parameters [169].

The importance of increasing the total area of greenery has already become a well-known fact [82]. However, the problem of limited undeveloped space in the city often does not allow the introduction of large green areas. Hence, attention should be paid to the possibility of introducing smaller urban green spaces [170], and their distribution in urban space should cause them to interact synergistically [22]. Kim et al. [171] proved that larger and well connected green areas contributed to the reduction of LST. In addition, less fragmented and ecologically healthy green areas were associated with lower temperatures.

### 3.4. Greenery along the Streets

It is suggested that the introduction of trees is the most effective and least costly way to mitigate UHI [25,92] and overall improvement of the quality of the urban environment [172]. Crowns of trees are used as a solution providing shade for the pedestrian zone urban canyon. They affect the reduction of temperature through evaporative cooling [163]. A review of the literature showed that the most frequently cited parameter for street greenery is the degree of shading of an urban canyon, and the LAI factor correlated with it. Shading reduces the average radiation temperature, which is one of the main parameters affecting the outside temperature. The shading effect can reduce the average summer radiation temperature by 30 °C [173]. Tree leaves transmit 10% visible and 30% solar infrared radiation and reflect 10% visible and 50% solar infrared radiation reaching the earth’s surface [103].

The amount of shadow cast by trees depends on the density of its crown [174,175]. The shape of the tree affects both the amount of shadow cast and the amount of radiation interception. The thick and dense tree crown can provide good quality shade, suggesting that deciduous trees are more effective than conifers [174,175,176,177,178,179]. Tree species with a lower crown temperature, such as *Populus nigra* L., *Quercus robur* L. or *Tilia cordata* Mill., are suitable for lowering the air temperature in an urban canyon [180]. Research results by Meier and Scherer [180] show that specific tree species can modulate the ambient temperature by specific values: *Populus nigra* L.—1.9 °C; *Quercus robur* L.—2.9 °C; *Fagus sylvatica* L.—3.2 °C; *Platanus × hispanica* Münchh.—3.9 °C; *Acer pseudoplatanus* L.—4.6 °C; *Acer platanoides* L.—5.0 °C, and *Acer campestre* L.—5.6 °C. *Tilia* sp. and *Corylus* sp. show the highest transpiration rates and the highest LAD according to studies conducted by Gillner et al. [181] and *Cinnamomum camphora* Ness et Eberm. and *Magnolia grandiflora* L. according to studies by Kong et al. [182]. Trees with wider crowns such as *Acacia confusa* Merr., *Macaranga tanarius* (L.) Müll.Arg. and *Ficus microcarpa* L.f. are more desirable in the space of the urban canyon than trees with smaller crowns such as *Melaleuca leucadendron* (L.) L. and *Livistona chinensis* (Jacq.) R.Br. ex Mart. [183]. It is also noted that the size of the leaves is not always an advantage. In the case when the trees do not have sufficient irrigation, the temperature of large leaves increases significantly, e.g., *Tilia platyphyllos* Scop. Small-leaved species such as *Gleditsia triacanthos* L. and *Metasequoia glyptostroboides* Hu & W. C. Cheng are recommended for places with extreme temperatures [177,182].

Optimization of planting patterns and distribution of trees is important in terms of both shading and heat release from under the tree crowns [15,145]. Depending on the location of trees in the urban canyon, weather conditions, and time of day, the trees may provide a cooling or warming effect [184]. A dense crown can be a problem due to heat retention at night [163]. In summer, however, and during extreme heat, trees play an essential role in mitigating the effect of the UHI [184]. The potential of a given species may remain unused if the tree is planted in the wrong place. The accuracy of tree location in an urban canyon seems to be as crucial as the morphological characteristics of the tree species. For example, trees with high LAI will be used most optimally when they are placed in open areas (squares or regions with low SVF) [184,185]. The planting patterns should also include a gap allowing ventilation of the system and the escape of long-wave radiation from under the crowns [166]. The joining of crowns at the same level, and thus blocking night radiation, may be prevented by planting trees of different forms [166,186].

When locating the tree, other parameters related to the characteristics of the environment should be considered. Studies by Park et al. [187] show greater efficiency in tree temperature mitigation if they were planted at the facades with a south-west exhibition. It is different from reservoirs located in an urban canyon. In this case, wind exposure seems significant. The results obtained by Syafii et al. [188] show that reservoirs oriented parallel to the main winds better maintain low radiation temperatures than other spatial configurations. When a reservoir is located parallel to the wind direction, temperatures drop by an average of 1.6–2.1 °C. The arrangement of ventilation corridors is also important for trees in dense building areas with low greenery available [189]. Ouyang et al. [190] studied the impact of the urban canyon green cover factor. The result was a non-linear distribution of values with a logarithmic formula. When the canyon coverage reaches 20–30%, the cooling effect remains almost unchanged. It is not fully understood how this result relates to water reservoirs, but generally larger reservoir (more water) bring more benefits in temperature mitigation [188].

### 3.5. Green Roofs

A green roof is a system consisting of layers that provides vegetation for plants. These layers are waterproofing, thermal insulation, vapor barrier, drainage, permeable soil substrate, and plant layer [191]. Green roofs are divided into extensive with thin substrates (2–20 cm) and their limited range of plants, and intensive, with a thicker layer of soil [192]. Roofs are considered to be one of the hottest surfaces in the city [193]. Many studies have proven green roofs reduce the temperature in their immediate surroundings by up to 10–15 °C [191] and increase the thermal insulation of buildings, thus reducing energy transfer to buildings and the need for cooling [22,92,191,192].

In the conducted query, the most frequently recurring quality parameter for green roofs was the degree of substrate hydration. The humidity of the green roof substrate depends on the intensity of evapotranspiration [194], as well as transpiration from the substrate. In the experiment by Ouldboukhitine et al. [195], transpiration depends on plants and substrate. Increasing the amount of water in the green roof substrate can reduce its thermal condensation [196]. The quality of the water used for irrigation is essential. The so-called gray water causes noticeable damage to plants, manifested as leaf curl and color change. Those damages decrease thermal resistance by up to 30% [195].

Despite the advancement of green roof designs, work is still underway to achieve a compromise on cooling performance with minimal irrigation and maintenance of plants on shallow soil substrates [197]. In this sense, the selection of plant species that increase thermal insulation in the system is essential [198]. In research by Azeñas et al. [198], significant differences were observed between the effectiveness of individual plant species. It is suggested that plant selection and leaf type seem to be more critical than transpiration. *Sedum sediforme* is more favorable in these studies than *Brachypodium phoenicoides*. For comparison, studies by Ouldboukhitine et al. [195] carried out on *Vinca minor* and *Lolium perenne* have shown the effect of significantly increasing the thermal resistance of the green roof after using periwinkle. The choice of plant species for green roofs, however, cannot be dictated by which species survive on the cheapest, shallow substrates. *Stachys* sp. and *Salvia* sp. were tested, with results surpassing other tested species in terms of surface and environment cooling, and with sufficient watering [199,200]. Plants other than *Sedum* sp. may play an essential role in easing UHI by green roofs, especially plants with bright leaf color, high albedo, and high LAI [200].

To effectively lower the temperature, green roofs must be overgrown with higher plants providing a high shading index and must be regularly irrigated [25,38,201]. Research by D’Orazio et al. [202] confirms that more important for blocking the energy flow into the building is a large leaf area and a high shading index than changing the substrate thickness. He et al. [203] studied the relationship between a substrate thickness and LAI. It turned out that the substrate thickness has a noticeable effect on the heat transfer through the roof and evaporation in both summer and winter, and the impact of LAI is significant in summer. After all, LAI is an important parameter that inversely affects the temperature fall [204]. It is assumed that when it comes to the efficiency of green roofs, those that are fully intensive are best, which means that the thickness of the substrate and the density of vegetation are two essential parameters affecting the temperature [15]. Interestingly, this is not always the case [205].

Particular attention should be paid to the correct location of green roofs. They should be placed on the possible lowest buildings and only where other BGI solutions cannot be used [25]. For high-rise buildings, the impact of the green roof on the surface temperature will be small [87]. The larger the roof, the smaller the LST and the larger the cool island buffer zone [206]. One green roof, however, does not significantly affect the surface temperature. This relationship can only be obtained by using a green roof system strategically located on a certain percentage of the buildings [25,207]. Research results by Dong et al. [206] show a decrease of 0.4 °C in LST for every 1000 m^2^ of green roof area growth.

### 3.6. Green Walls and Facades

Green walls are objects consisting of structures integrated with the walls of buildings and plants planted in unique vegetation systems. Similar are green facades, but they consist of plants planted in the ground or in containers and growing on supporting structures or directly on the walls of buildings. Like green roofs, green walls are promoted because of their insulating properties and lowering the ambient temperature through evapotranspiration [22,119,208,209]. Green walls and facades reduce the UHI effect in both summer and winter. In winter, they limit the emission of anthropogenic heat [103]. Due to the low installation and maintenance costs, green facades are recommended more often than green walls when implementing BGI [210]. The literature also describes active green walls with forced air circulation. Despite frequent installation of such elements on the facade, they mainly affect the internal environment of the building [211].

The intensity of cooling through green walls is most strongly related to the building’s orientation relative to the world, and thus the level of exposure to solar radiation. Walls with high sunlight should be chosen, especially when limited space does not allow the introduction of woody plants, and other BGI solutions cannot be used [103]. In the northern hemisphere (southern orientation, and the southern hemisphere), northern orientation should be used to provide sufficient energy for effective evapotranspiration cooling [212]. In these configurations, the wall is more exposed to solar radiation [213]. The effect of lowering the temperature depends on the geographical location and the season of year or time of day. In some places on Earth, western orientation will bring the most satisfactory results [214], and depending on the time of day it may be the eastern side [213]. Exposure to the sun is variable. Green wall systems should be adapted to local conditions anywhere in the world since it is impractical to use them on each wall of the building [215]. On the other hand, the greening of less favorable walls from the evaporative cooling point of view can compensate for their weak, passive construction, and effectively reduce the need for cooling inside the building. In this case, the most effective is the western or eastern wall [216]. The color of the walls is essential from a temperature point-of-view. Dark-colored walls are designated as necessary for covering with vegetation because they heat up more than walls with high albedo [25,216].

Several other parameters have a significant impact on the thermal capacity of the green wall [217]. Research results show that the LAI increase from 1 to 5 can reduce the surface temperature by 12 °C on a hot summer day, and reduce the annual cost of indoor air conditioning by 1.4% [218]. Larger leaves provide better cooling in every spatial configuration of the green wall [216,219]. Besides, the most significant temperature drops are observed in places where the vegetation layer is thicker [220]. An experiment carried out by Koyama et al. [221] revealed that a leaf solar radiation permeability determines the reduction of the temperature measured on the wall. Thicker leaf blades can delay the rise of ambient temperature in the morning by preventing surface heating [213].

The effectiveness of green walls and facades in lowering the ambient temperature depends on irrigation [103]. The quality of evapotranspiration depends on the health status of plants, which can only be ensured by balanced watering. Designers should, therefore, be aware of the difference in vertical moisture distribution when creating automated systems [222]. The type of vegetation and the type of green wall technology are essential as well [103]. Depending on the purpose the designer wants to achieve, direct or indirect facade greening should be chosen [223]. Tudiwer and Korjenic [224], as well as Yin et al. [225], showed that double-layer green walls have a significant impact on cooling performance. Kontoleon and Eumorfopoulou [216] claim the insulation of building walls additionally stabilizes temperature fluctuations. Lee and Jim [212] argue that enlargement of the air gap between the green wall and the building can lead to lowering the ambient temperature. When designing green walls, special attention should be given to the place where they are introduced. The percentage of walls greenery should increase as the density of city buildings decreases [215], which can be explained by a greater degree of exposure to solar radiation. The most important, however, is the precise placement of individual solutions in the urban tissue to achieve the best thermal effects. The results presented by Morakinyo et al. [215] have shown that it is possible to reduce the local temperature in the city by 1 °C during day and night, with 30–50% greening of the facade walls in a densely built-up city.

## 4. Conclusions and Recommendations for Further Research

Our review paper presents a systematic review of BGI quality parameters focused on UHI mitigation and exclusively dependent on designers (Table A1). The results of the literature analysis showed the most critical geometrical, morphological, and topographic parameters. The classified parameters concerned five types of elements included in the blue-green infrastructure. For water objects and greenery along the streets, the most important family of parameters are topographic parameters, depending on the spatial context. For green areas, green roofs, and green walls, these are morphological parameters, depending on the characteristics of the object itself. Important parameters from the UHI mitigation point-of-view are surface for water facilities and green areas, leaf size/LAI for greenery along the streets, hydration level/substrate moisture for green roofs, and orientation of green walls relative to the directions of the world.

Creating cityscapes using computer parametric modeling is a relatively new approach. Parametric modeling is a process with the ability to reshape geometry when the dimension value changes. This process is automated using an algorithm where dimensions and shapes are defined, but may be changed using the input data [226]. The results obtained in this bibliographic study may serve as guidelines for obtaining further numerical input data, and the main goal of the authors is to create a site specific BGI modeling algorithm aimed at mitigating UHI.

In the literature, one can find a multitude of parameters of all BGI elements and specific values of their impact on the ambient temperature. Unfortunately, greenery is often overlooked as an element of the infrastructure that covers the entire city, and the effect of individual BGI elements on each other is rarely studied. Few scientific articles describe the synergy effect that can occur between elements of infrastructure [215,227].

It is the synergistic cooperation in influencing UHI that should become a new research direction related to climate change mitigation in the city.

In creating a holistic model of BGI for the UHI mitigation, one must consider that each city is unique, and therefore site-specific UHI models usually perform best by predicting the development of this atmospheric phenomenon, especially under the climate change perspective. Finding and describing general rules for UHI mitigation in the temperate climate (C) zone is not a simple task. Many different factors must be considered to reach this goal. For example, different weather types completely change the UHI intensity and factors determining it. If climate change brings more advection, UHI intensity and morphology will change, and other mitigation measures must be considered.

## Figures and Tables

**Figure 1 ijerph-17-07093-f001:**
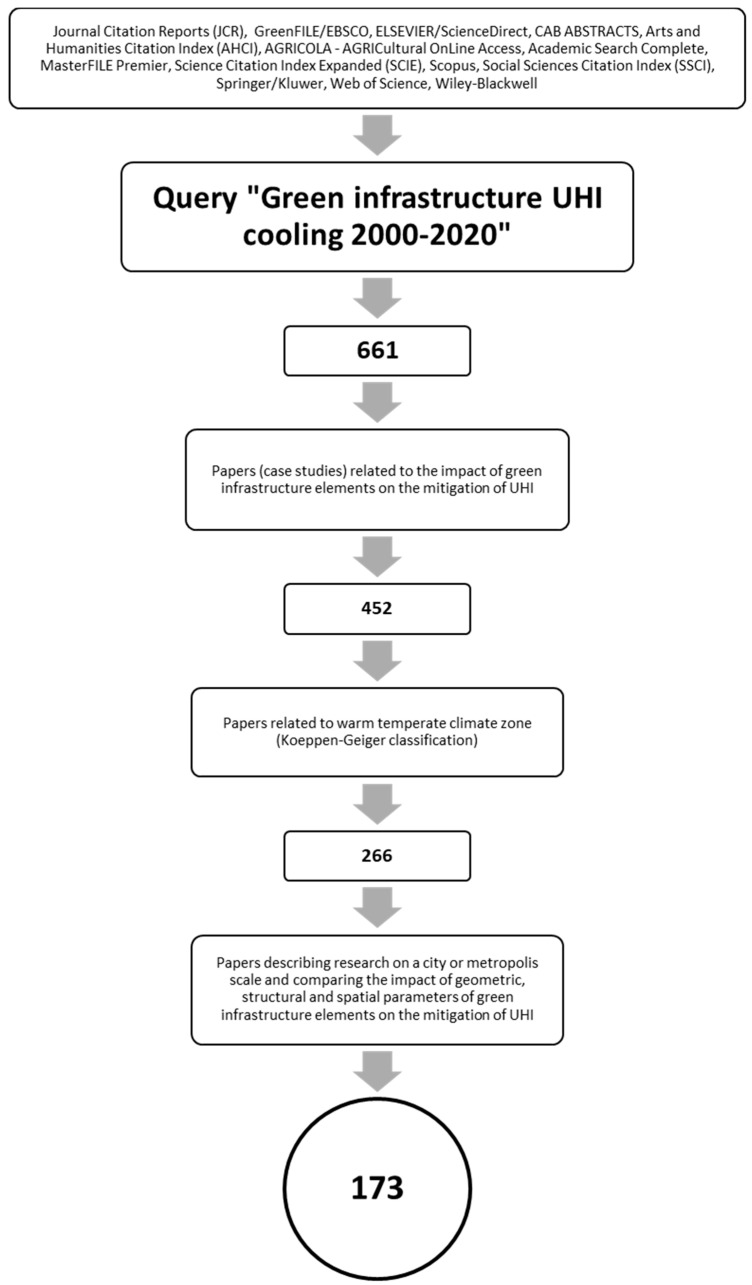
Logical diagram of papers selected for research.

**Figure 2 ijerph-17-07093-f002:**
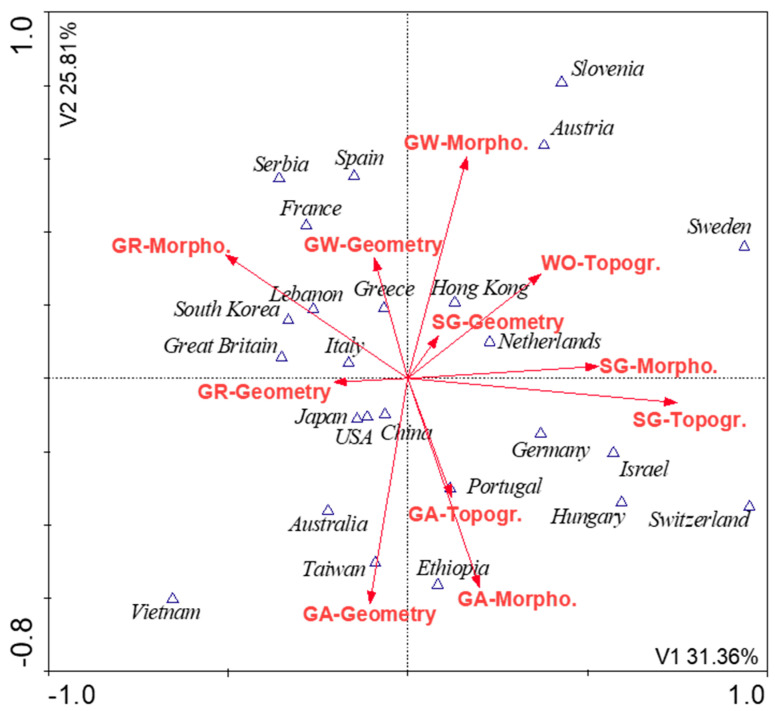
The Canonical Conformity Analysis (CCA) analysis (n = 167). Relationships between the frequency of research on parameter families in selected countries based on papers included in the analyses. N publications per country of origin (Hong Kong 27; China 25; Great Britain 13; Japan 12; Australia 11; Germany 11; USA 9; Italy 8; Israel 8; Netherlands 7; Greece 6; Other <5).

**Table 1 ijerph-17-07093-t001:** Parameters affecting Urban Heat Island (UHI) intensity that depend on the decision-making process [14,26,29,30,31,32,33,37,39,41,42,43,44,45,46,47,48,49,50,51,52,53,54,55,56].

Parameter Group	Parameter
Related to surface properties	Heat capacity Short-wave radiation and long-wave radiation absorption Degree of evaporative cooling Degree of transpiration
Related to the type of building materials	Albedo of building materials Radiation from building surfaces and infrastructure Heat loss in buildings Surface permeability in the city
Related to geometry and topography	Sky visibility factor Canyons proportions Length of rough surfaces Building geometry Building density Geometry of the city’s spatial arrangement Characteristics of the spatial arrangement
Related to the type of land cover	The number of biologically active areas and green areas The presence of water reservoirs and rivers Extent of sealed surfaces Increased wind flow and speed
Related to living and managing	Use of buildings Anthropogenic heat release Water management Change in use land

**Table 2 ijerph-17-07093-t002:** Earlier review papers addressing the impact of blue-green infrastructure (BGI) parameters on UHI mitigation.

Author	Title	Usefulness for Creating Design Guidelines
[98]	Critical review on the cooling effect of urban blue-green space: A threshold-size perspective	Lists many parameters for water elements and green areas. Discusses the synergy between a water element and a green area
[99]	Green roofs to reduce building energy use? A review on key structural factors of green roofs and their effects on urban climate	Lists green roofs parameters
[100]	A review of mitigating strategies to improve the thermal environment and thermal comfort in urban outdoor spaces	Lists green areas and water elements parameters.
[101]	Urban green space cooling effect in cities	Detailed analysis of the impact of the size of green areas on UHI. Description of possible synergies between green areas
[102]	The evapotranspiration process in green roofs: A review	Lists green roofs parameters
[103]	Chapter 19-WSUD and Urban Heat Island Effect Mitigation	Comprehensive description of water elements, street greenery, green areas, green roofs, and green walls parameters
[104]	Outdoor thermal comfort by different heat mitigation strategies-A review	Lists green areas, street greenery, green roofs, and green walls parameters
[105]	Green roofs and facades: A comprehensive review	Detailed description of green roofs, green walls, and green facades parameters
[106]	The impact of urban compactness, comfort strategies and energy consumption on tropical urban heat island intensity: A review	Lists green roofs, green walls, and green areas parameters
[107]	Vertical greenery systems: A systematic review of research trends	Extensive description for green walls and green facades parameters
[108]	Evaluating the cooling effects of green infrastructure: A systematic review of methods, indicators, and data sources	Very general but contains many case studies descriptions
[109]	Progress in urban greenery mitigation science–assessment Methodologies advanced technologies and impact on Cities	Thoroughly described green areas parameters. Discussion on the green areas synergy effect
[110]	Approaches to Outdoor Thermal Comfort Thresholds through Public Space Design: A Review	Lists street greenery parameters
[111]	How to Design a Park and Its Surrounding Urban Morphology to Optimize the Spreading of Cool Air?	Comprehensive description of green areas parameters
[112]	Vertical greening systems–A review on recent technologies and research advancement	Detailed description of green walls and facades parameters
[113]	Current trends in urban heat island mitigation research: Observations based on a comprehensive research repository	Lists many water elements, street greenery, and green areas parameters
[114]	Energy conservation and renewable technologies for buildings to face the impact of the climate change and minimize the use of cooling	Lists green roofs and green walls parameters. Describes water roofs.
[115]	Heat mitigation by greening the cities, a review study	In-depth analysis of green areas parameters
[11]	Review on the impact of urban geometry and pedestrian level greening on outdoor thermal comfort	The impact of Sky View Factor as well as street greenery and city parks parameters is widely described
[61]	Regulating the damaged thermostat of the cities–Status, impacts and mitigation challenges	Lists green areas and green roofs parameters
[116]	Green infrastructure as life support: urban nature and climate change	Lists green areas parameters
[25]	Planning for cooler cities: A framework to prioritize green infrastructure to mitigate high temperatures in urban landscapes	Lists parameters for street greenery, city parks, green facades, and green roofs
[117]	Responses of tree species to heat waves and extreme heat events	Detailed description of parameters for trees
[118]	Vertical Greenery Systems (VGS) for energy saving in buildings: A review	Extensive parameter description for green walls and green facades
[71]	Cooling the cities–A review of reflective and green roof mitigation technologies to fight heat island and improve comfort in urban environments	Detailed comparison of the green roofs parameters
[119]	Quantifying the thermal performance of green façades: A critical review	Detailed description of green facades
[120]	REVIEW Effects of Evapotranspiration on Mitigation of Urban Temperature by Vegetation and Urban Agriculture	Parameters for green areas, green roofs, and water facilities
[81]	Evidence for the temperature-mitigating capacity of urban blue space–a health geographic perspective	Widely described water elements parameters
[121]	Performance evaluation and development strategies for green roofs in Taiwan: A review	Lists parameters for green areas, street greenery, and green roofs
[82]	Urban greening to cool towns and cities: A systematic review of the empirical evidence	Lists green roofs parameters
[122]	The International Urban Energy Balance Models Comparison Project: First Results from Phase 1	Lists green areas parameters

Note: Reference numbers match the reference numbers in main text.

**Table 3 ijerph-17-07093-t003:** Classification of the significance of design parameters of BGI elements affecting UHI mitigation in a warm temperate climate zone.

BGI Structure	Parameter Family	Dominant Parameter	Number of Papers	Percentage
Water structures			47	100
	Geometrical parameters		15	31.92
		Area	9	-
	Morphological parameters		4	8.51
		The degree of vegetation along the banks	2	-
	Topographic parameters		28	59.57
		Wind exposure (trend of cold transfer)	6	-
Green areas			231	100
	Geometrical parameters		51	22.08
		Area	32	-
	Morphological parameters		125	54.11
		Percentage of an area covered by trees	14	-
	Topographic parameters		55	23.81
		Exposure to solar radiation/degree of shading of the area by surrounding structures	6	-
Greenery along the streets			110	100
	Geometrical parameters		26	23.48
		Tree crown width/tree crown diameter	8	-
	Morphological parameters		39	35.65
		Leaf size/LAI	11	-
	Topographic parameters		45	40.87
		Canyon geometry	5	-
		Canyon Height	5	-
		Canyon Width	5	-
Green roofs			121	100
	Geometrical parameters		20	16.80
		Substrate layer thickness	10	-
	Morphological parameters		92	76.80
		Degree of hydration/moisture of the substrate	24	-
	Topographic parameters		8	6.40
		The height of the structure above the ground	2	-
		Distance from the nearest neighboring BGI (synergy)	2	-
Green walls			59	100
	Geometrical parameters		15	25.42
		Degree of vegetation coverage of a building/the extent of the green wall	8	-
	Morphological parameters		33	55.93
		Leaf width, leaf area, foliage density/LAI	8	-
	Topographic parameters		11	18.65
		Location relative to the directions of the world	9	-

**Table 4 ijerph-17-07093-t004:** Statistical parameters for the Canonical Conformity Analysis (CCA) analysis presented in Figure 2.

Number of variables	15		
Number of rejected variables	4		
Number of permutations	9999		
**Parameter family**	***p*-Value**	**F-Value**	**% Expl.**
GW-Morpho.	0.001	16.265	14.25
SG-Topogr.	0.001	14.632	11.14
GA-Morpho.	0.001	14.023	10.53
GA-Geometry	0.001	13.269	9.21
GR-Morpho.	0.001	10.961	8.36
WO-Topogr.	0.001	9.863	7.77
SG-Morpho.	0.002	9.569	7.54
GA-Topogr.	0.003	6.991	6.87
GW-Geometry	0.007	6.657	6.42
GR-Geometry	0.018	4.269	5.26
SG-Geometry	0.036	2.539	4.98

GW-Morpho.—Green walls-morphological parameters; SG-Topogr.—Greenery along the streets-topographic parameters; GA-Morpho.—Green areas-morphological parameters; GA-Geometry—Green areas-geometrical parameters; GR-Morpho.—Green roofs-morphological parameters; WO-Topogr.—Water structures-topographic parameters; SG-Morpho.—Greenery along the streets-morphological parameters; GA-Topogr.—Green areas-topographic parameters; GW-Geometry—Green walls-geometrical parameters; GR-Geometry—Green roofs-geometrical parameters; SG-Geometry—Greenery along the streets-geometrical parameters.

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
