# Peer review of "Statistical Review of Quality Parameters of Blue-Green Infrastructure Elements Important in Mitigating the Effect of the Urban Heat Island in the Temperate Climate (C) Zone"

_ijerph, 2020, doi:10.3390/ijerph17197093_

Round 1

Reviewer 1 Report

The article is well written. And uses a novel statistical approach to describe geographical research trends across blue green infrastructure types. The article is well written, and I think a good contribution to the journal. The findings are relatively low impact, consiering the vast amount of literature used - and I think the discussion could be refined to be more impactful. Most of the discussion does not discuss the results, but merely forms additional literature review which could be placed in the introduction.

Some recomendations are as follows:

CVA acronym needs to be explained in the abstract. 

CCA is not widely utilised across all disciplines that stand to gain from reading this MS, so some additional information would be advantageous in the interpretation of Figure 2 -  

I would like to see some discussion on active living wall technology for the provision of temperature modulation:

The discussion could benefit from having additional commentary on BGI policy, and one way to increase the impact of this work is to link manuscript frequency per country with relevant policies for each of those places.

Reviewer 2 Report

The manuscript entitled "Review of quality parameters of Blue-Green Infrastructure elements important in mitigating the effect of the Urban Heat Island in a moderate warm climate (first stage of creating site specific BGI modeling script)" presents a systematic review of essential BGI objects for UHI mitigation. The article presents a topic of great relevance and importance for urban climate studies, however some details must be corrected. - The title is very long and confusing, I just suggest: "Review of quality parameters of Blue-Green Infrastructure elements important in mitigating the effect of the Urban Heat Island". - Abstract: the first sentence is out of context, I recommend removing it because the work is not about climate change but about a common phenomenon in urban areas, the UHI. Authors should make clear the purpose of the research and the methods used to prepare the systematic review in a summarized form. - The introduction is timid and does not consistently introduce the research theme. It is necessary to enrich the introduction with more references and comments about UHI and BGI. - The last paragraph of item 1.2, "This paper aims to systematically present geometric, structural, and spatial parameters of BGI elements that may affect mitigating the effect of the UHI in a temperate climate zone. This paper is also the first in a series of articles describing the creation of a dynamic algorithm for thermal optimization of cities in the temperate climate zone using BGI solutions." I suggest moving it to the end of the introduction. - In the last sentence of the second paragraph of the methodology: "Hence in obtaining data for this paper, it was decided to focus on only one climate zone." define the chosen climate zone immediately. -In item 2.1 of the methodology: describe in detail which methods were used for the development of each of the 4 stages. - It is not clear what was the base question of the systematic review, as well as the keywords used for the search. -In the clusion of the research, point out the gaps observed in the work and the possibilities of future work that can solve them. After making the corrections pointed out, I recommend the publication of the manuscript.

Reviewer 3 Report

Comments on the ijerph-927401-peer-review-v1 ”Review of quality parameters of Blue-Green Infrastructure elements important in mitigating the effect of the Urban Heat Island in a moderate warm climate (first stage of creating site specific BGI modeling script)” by Patryk Antoszewski, Dariusz Åšwierk and MichaÅ‚ Krzyżaniak

The manuscript ”Review of quality parameters of Blue-Green Infrastructure elements important in mitigating the effect of the Urban Heat Island in a moderate warm climate (first stage of creating site specific BGI modeling script)” by Patryk Antoszewski, Dariusz Åšwierk and MichaÅ‚ Krzyżaniak tried to summarize the impact of the Blue-Green Infrastructure (BGI) elements to mitigate the effect of Urban Heat Island (UHI) in a moderate warm climate through a literature data analysis based on the statistical model (CVA). It is found that the area of water objects and green areas, street greenery leaf size (LAI), green roofs hydration degree, and the green walls location of the BGI objects parameters are essential for the UHI mitigation.

Specific points

  1. The title of the paper is too long and does not focus on the theme of the manuscript’s context.
  2. Please specify how the authors applying the CCA analysis to the BGI elements to mitigate the UHI effect ? Please give further explanations to Figure 2.

Reviewer 4 Report

General comment

Detailed and very complete bibliographical work. The introduction (both phenomenological and global) and the analysis considering each type of solution are relevant and interesting. Most of the main important features are well reviewed.

4 parts :

  • Introduction (very clear: two sub parts : the effect and the BGI solutions )
  • Material and Methods: important task to introduce judicious classifications and to appreciate the most studied points as well as the gaps: this chapter seems complete and well done
  • Result and discussion: The main results concerning the BGI are well identified and well commented. References are well provided and relevant
  • Conclusion and recommendations: one would expect new openings here on databases (described as very insufficient) and on design tools as numerical modeling. Some references talk about, but it should be introduced here: is this the next step?

Detailed comments

  • Table 1 : This list is interesting but it seems a little messy. We recommend classifying it for example according to the targeted underlying heat exchange process: Storage and conduction, Radiation, latent heat of phase change, convection ...
  • Line 165 : List of earlier review papers : A new column with the comment of the authors should be appreciated (what is estimated as good or not)
  • The cascade of successive filters could be commented: why 4 filters? why this order?
  • Figure 2 deserves more comments: We seem to understand the different centres of interest that each country has or are developing but we do not find a climate logic? It would probably be interesting to include the number of publications per country considered in this study.

Round 2

Reviewer 3 Report

The authors addressed my concerns related to the manuscript. I have no further questions and comments on the paper.